# Putting the microbiota to work: Epigenetic effects of early life antibiotic treatment are associated with immune-related pathways and reduced epithelial necrosis following *Salmonella* Typhimurium challenge *in vitro*

**Matheus O. Costa** [1,2]*, **Janelle Fouhse**[3], **Ana Paula P. Silva**[4], **Benjamin Willing**[3], **John C. S. Harding**[1]

**1** Large Animal Clinical Sciences, Western College of Veterinary Medicine, University of Saskatchewan, Saskatoon, Saskatchewan, Canada, **2** Department of Population Health Sciences, Faculty of Veterinary Medicine, Utrecht University, Utrecht, The Netherlands, **3** Department of Agricultural Food and Nutritional Science, University of Alberta, Edmonton, Alberta, Canada, **4** Faculdade de Veterinária, Universidade Federal do Rio Grande do Sul, Porto Alegre, Brazil

* matheus.costa@usask.ca

**Data Availability Statement:** Metagenomic data was uploaded to ENA (PRJEB33426), and Transcriptomic data to GEO GSE134130

## Abstract

*Salmonella enterica* serovar Typhimurium is an animal welfare and public health concern due to its ability to parasite livestock and potentially contaminate pork products. To reduce *Salmonella* shedding and the risk of pork contamination, antibiotic therapy is used and can contribute to antimicrobial resistance. Here we hypothesized that immune system education by the microbiota can play a role in intestinal resilience to infection. We used amoxicillin (15mg/Kg) to modulate the intestinal microbiome of 10 piglets, paired with same age pigs that received a placebo (n = 10) from 0 to 14 days of age. Animals were euthanized at 4-weeks old. Each pig donated colon sections for *ex vivo* culture (n = 20 explants/pig). Explants were inoculated with *S.* Typhimurium, PBS or LPS (n = 6 explants/pig/group, plus technical controls). The gut bacteriome was characterized by sequencing of the 16S rRNA at 7, 21 days of age, and upon *in vitro* culture. Explants response to infection was profiled through high-throughput mRNA sequencing. *In* vivo antibiotic treatment led to β-diversity differences between groups at all times (*P*<0.05), while α-diversity did not differ between amoxicillin and placebo groups on day 21 and at euthanasia (*P*<0.03 on day 7). Explant microbiomes were not different from *in vivo*. *In vitro* challenge with *S.* Typhimurium led to lower necrosis scores in explants from amoxicillin-treated pigs, when compared to explants placebo-treated pigs (*P*<0.05). This was coupled with the activation of immune-related pathways in explants from amoxicillin-treated pigs (IL-2 production, NO production, BCR activation), when compared to placebo-treated pigs. In addition, several DNA repair and intestinal wound healing pathways were also only activated in explants from amoxicillin-treated pigs. Taken together, these findings suggest that immune education by the amoxicillin-disturbed microbiota may have contributed to mitigate intestinal lesions following pathogen exposure.

**Funding:** JCSH - Grant number 345169 - 2014 Large-Scale Applied Research Project Competition – Genomics and Feeding the Future - https://www.genomecanada.ca/en/programs/large-scale-science/past-competitions/large-scale-research-project-competitions/2014-large The funders had no role in study design, data collection and analysis, decision to publish, or preparation of the manuscript.

**Competing interests:** The authors have declared that no competing interests exist.

## Introduction

*Salmonella enterica* serovar Typhimurium has a broad range of host tropism that includes swine and humans, leading to gastrointestinal disease in both species. It is an important cause of food-borne zoonoses worldwide, being associated with millions of cases of gastroenteritis and bacteremia in humans every year [1]. Pigs are an important infection source for humans, as they can shed the pathogen without clinical signs or become chronic carriers, increasing the risk of carcass contamination [2, 3]. *S. enterica* serovars lead to fibrinonecrotic enterocolitis, diarrhea and dehydration in pigs. This disease is commonly found in animals with concurrent debilitating infections, or if raised in environments that facilitate exposure to high doses of the pathogen, such as modern intensive rearing systems [4, 5]. While avirulent pig vaccines for *S. enterica* serovars are available, vaccination is not recommended in pigs as carriers can still shed the bacteria and infect humans [6, 7]. Prevention and control are mainly based on disinfection of premises, biosecurity measures and mass treatment with antimicrobial agents. Both metaphylactic and prophylactic strategies have been applied, resulting in selective pressure that potentially contributes to the emergence of antibiotic resistant strains [8]. In fact, *S.* Typhimurium is one of the few serovars in which multi-drug resistance has been documented, specifically through the emergence of strain DT104 world-wide [9–12]. In face of this challenge, a quest for non-antimicrobial alternatives to treat and prevent *Salmonella* infection and colonization in pigs has launched. There has been growing interest in the intestinal microbiota as a potential tool to prevent disease and colonization, as it is suggested that *S.* Typhimurium exploits the host intestinal inflammation to overcome the indigenous microbiota [13].

The use of antibiotics in pigs and mice has been observed to cause a characteristic bloom of Enterobacteriaceae (more specifically *Escherichia coli*) following parenteral delivery [14, 15]. Other research has recently suggested that microbiota-derived LPS may be one mechanism of immune education, with *E. coli* LPS stimulating the immune system [16]. Current work by our group has shown that amoxicillin treatment of piglets has long term effects on the local and system immune response (Fouhse et al., submitted). Clarifying if this antibiotic-induced microbiota disruption can improve resilience to intestinal infections is the first step towards the development of non-antibiotic based strategies to shift the composition of microbial populations that recapitulate this immune enhancement.

Here we hypothesized that modulation of the swine intestinal microbiome preceding exposure to *S.* Typhimurium can minimize intestinal lesions upon challenge. We explored this theory using a combination of *in vitro* and *in vivo* models and high-throughput mRNA sequencing.

## Material and methods

Animal experiments were designed and conducted in accordance with the Canadian Council for Animal Care and approved by the University of Saskatchewan Committee on Animal Care and Supply (Protocol #20170044).

### *In vivo* procedures

Groups of 4 piglets from 5 different litters (n = 20, born within 24 hours) were enrolled in this study. Dams were parity 2 (n = 3), 4 (n = 1) and 5 (n = 1). Piglets were purchased from a PRRS negative, high-health commercial herd, with no recent history of diarrheic disease or diagnosis of Salmonellosis. At birth, piglets were paired within litter by gender and birth weight, and randomly assigned to one of the following treatment groups: amoxicillin (n = 10, Apotex, Toronto, Canada, PO, sid, 15 mg/Kg) or placebo (PBS; phosphate-buffered saline, pH 7.4, n = 10, PO, sid, same volume as amoxicillin/weight). Treatments began on day 0, defined as

the day when the oldest piglet from a given litter was less than 24 hours old. Treatment dose was adjusted for body weight at birth and corrected every 72 hours until 14 days after farrowing. At 4–5 weeks of age, piglets were transported to the research facility and immediately euthanized upon arrival by captive bolt followed by exsanguination. To simplify the workflow, pigs were euthanized in batches of 4 per day for 5 consecutive days, resulting in age at euthanasia ranging from 25 to 28 days. Piglet weights, treatment doses and age at euthanasia are provided in the S1 Table.

## *In vitro* colon culture and inoculation

Tissue collection and preparation for culture was performed as previously described [17, 18]. The following procedures were repeated for each pig. Immediately after euthanasia, a 10 cm segment of spiral colon (apex) was excised and placed in 50 mL of Dulbecco phosphate buffered saline (DPBS, without $Ca^{2+}$ and $Mg^{2+}$, 0.1 M, pH 7.0). Within one hour of euthanasia, the colonic serosa was separated from the mucosa on a refrigerated surface while embedded in DPBS. The mucosa was divided into 1.5 cm x 1.5 cm explants (n = 20/pig), which were placed individually on cell strainers (70 μm mesh, Corning, Corning, NY, USA) in 6-well plates, luminal surface facing up. Each well received 3 mL of culture media (KBM-gold Keratinocyte, Lonza, Walkersville, USA) supplemented with 1.5 mM $Ca^{2+}$. A sterile polypropylene inoculation ring was attached to the mucosa of each explant using medical-grade adhesive (3M Vetbond Tissue Adhesive, 3M Animal Care products, St. Paul, MN, USA). Explants were randomly allocated into 3 groups, with each explant receiving 100 μL of inoculum: i) negative control (inoculated with sterile PBS, n = 6), ii) positive control (inoculated with 100 μg of LPS, *E. coli* O:127 B:8, Sigma, St. Louis, USA, n = 6), or iii) *Salmonella enterica* serovar Typhimurium strain SL1344 (n = 6). Plates were transferred into modular chambers (Billups-Rothenberg Inc., Del Mar, CA, USA) infused with 95% $O_2$ + 5% $CO_2$ at 37˚C. After 0 (immediately after being placed in the incubator) and 30 minutes in culture, explants were removed from the incubator and immersed in formalin (n = 3/group) or RNAlater (n = 3/group, Qiagen, Mississauga, ON, Canada).

## Inoculum preparation and swab screening for *Salmonella* spp

*Salmonella enterica* serovar Typhimurium strain X4232 was cultured following standard culture procedures: overnight at 37˚C with shaking (200 rpm) in Luria-Bertani (LB, BD Canada, Oakville, ON, Canada) broth. An aliquot was seeded in sterile LB broth 2 hours before inoculation of explants, growth was observed using optical density (600 nm, Vmax microplate reader, Molecular Devices, San Jose, CA, USA) in order to have log-phase, fresh cultures used in as inoculum. After explant setup for culture, a 10μL aliquot of the inoculum was spread on XLT-4 agar (xylose-lysine-tergitol 4, BD Canada, Oakville, ON), and tested for viability by incubating plates for 24 hours at 37˚C. Samples were considered positive for *Salmonella* spp. if black or red colonies with black centres were observed (due to the reduction of thiosulphate to hydrogen sulphide by the bacterium). All inocula were positive for *Salmonella*-like colonies, and explants were exposed to an average of 5.36 x $10^8$ cfu/mL, with no daily inoculum having less than 2 x $10^8$ cfu/mL. Furthermore, all pigs (n = 20) were tested for *Salmonella* spp. colonization. Rectal swabs collected immediately prior to euthanasia were plated on XLT-4 agar and cultured for 24 hours at 37˚C. In addition, 5 μL of inoculum collected from explants after the 30 minutes incubation period were tested for the presence of *Salmonella*. All rectal swabs tested negative, and all inocula tested positive, according to the plate-reading standards described above.

## Histopathology and slide scoring

Formalin fixed explant sections were stained with hematoxylin and eosin (H&E) as per standard protocols. Histopathological analysis was performed by an observer (MC), blinded to sample identity, using optical microscopy at 20× magnification. Necrosis scores were based on the evaluation of epithelial cells within all crypts and the surface epithelium of each explant for which the entire crypt length was visible. Necrotic/apoptotic cells were characterized by reduction or gain in cellular volume, loss of characteristic columnar format and DNA fragmentation (score 0: no necrotic cells visible, score 1: <10% necrotic cells, score 2: 11%–35% necrotic cells, score 3: 36–70% necrotic cells, score 4: >71% of cells are necrotic). Average necrosis scores from H&E stained explant sections were compared using generalized estimating equations (GEE) using an unstructured correlated working matrix while clustering by pig (SPSS v21, IBM, New York, NY, USA).

## Fecal and mucosa swab collection and 16S rRNA gene amplicon sequencing

Rectal swabs used for 16S rRNA sequencing were collected from all pigs (n = 20/group) on 7 and 21 days post-farrowing. In addition, the colonic mucosa used as explant source from each pig was swabbed after euthanasia (pre-washing) and immediately before explants were placed in the incubator (post-washing).

DNA extraction and 16S ribosomal RNA (rRNA) sequencing procedures followed previously published methodologies [19]. In summary, total DNA from swabs was extracted using QIAamp DNA Mini Stool Kit (Qiagen Inc., Valencia, CA, USA) following guidelines by the manufacturer together with a bead-beating step (FastPrep instrument; MP Biomedicals, Solon, OH, USA). DNA concentration was determined using Quant-iT™ PicoGreen® dsDNA Assay Kit (Thermo Fisher Scientific, Waltham, MA, USA). DNA (5 ng/μl) was amplified targeting V3-V4 regions of the bacterial 16S rRNA gene with universal primers using KAPA HiFidelity Hot Start Polymerase (Kapa Biosystems Inc., Wilmington, MA, USA). PCR cycling conditions for 16S rRNA amplification were: 5 min at 95˚C, 25 cycles of 20 s at 98˚C, 15 s at 55˚C, 1 m at 72˚C, hold at 4˚C. Subsequently, PCR products were purified using AMPure XP beads (Beckman Coulter Inc., Mississauga, ON, Canada), then dual indices and Illumina sequencing adapters were attached using Nextera XT Index Kit (Illumina Inc., Victoria, BC, Canada). Cycling conditions were 5 min at 95˚C, 10 cycles of 20 s 98˚C, 15 s 55˚C, 1 min 72˚C, hold at 4˚C. PCR products were purified and diluted to 4 nM. Aliquots of the 4 nM products were pooled, size-selected, denatured with NaOH, diluted to 4 pM in Illumina HT1 buffer, spiked with 10% PhiX, and heat denatured at 96˚C for 2 min before loading. A MiSeq 600 cycle v3 kit was used to sequence each sample (Illumina MiSeq). Nextera adapter sequences were used for run trimming.

## Transcriptome sequencing

The LPS (*in vitro* positive control) group was excluded from this analysis. Total RNA was extracted from fixed colon explants using a commercial kit (Qiagen RNeasy, Mini Kit (QIAGEN, Mississauga, ON, Canada) following the manufacturer's instructions. Extracted RNA integrity was assessed using the Agilent 2100 Bioanalyzer (Agilent Technologies, Santa Clara, CA, USA) and Nanodrop 2100 spectrophotometer (Thermo Scientific, Wilmington, DE, USA) was used to measure the concentration and investigate the presence of contaminants. rRNA were depleted from 400 ng of total RNA using Ribo-Zero™ rRNA Removal Kits (Meta-Bacteria, Epicentre, Madison, WI, USA. Residual RNA was cleaned up using the Agencourt RNACleanTM XP Kit (Beckman Coulter, Indianapolis, IN, USA) and eluted in water. cDNA synthesis was achieved with the NEBNext RNA First Strand Synthesis and NEBNext Ultra Directional RNA Second Strand Synthesis Modules (New England Biolabs, Pickering, ON,

Canada). The remaining steps of library preparation were done using and the NEBNext Ultra II DNA Library Prep Kit for Illumina (New England Biolabs, Pickering, ON, Canada). Final libraries were quantified using the Quant-iT™ PicoGreen® dsDNA Assay Kit (Life Technologies, Burlington, ON, Canada) and the Kapa Illumina GA with Revised Primers-SYBR Fast Universal kit (Kapa Biosystems Wilmington, MA, USA). Average size fragment was determined using a LabChip GX (PerkinElmer, Hopkinton, MA, USA) instrument. Individually indexed libraries were sequenced in four lanes on the Illumina HiSeq 2000 system at the McGill University, Genome Quebec Innovation Centre (Quebec, Canada) to obtain high-quality, 100-bp paired-end reads (average phred quality score $\geq$ 36).

## Bioinformatic analyses

**16S rRNA gene amplicon sequencing processing.** Data analysis was performed by an individual blinded to *in vivo* sample identity (JF). Raw sequence data was processed using the default on-rig procedures from Illumina [20]. Raw sequence reads were merged using PANDAseq and downstream analysis was performed using QIIME (1.9.1, [21]). Chimeras were removed using a UCHIME and UPARSE workflow and resulting sequences were clustered into operational taxonomic units (OTUs) having > 97% similarity with USEARCH [22, 23]. Taxonomy was assigned using Ribosomal Database Project classifier V2 [24]. Shannon and Chao1 indices were used to estimate alpha diversity using the phyloseq package (v1.22.3) in R (v.3.5.3). Phyloseq was also used to visualize changes to microbial community structure using the Bray Curtis dissimilarity and principal-coordinate analysis (PCoA) and analysis of similarities (ANOSIM) was used to test differences between treatment groups. Differential abundance of dominant taxa at the phylum and genus levels were compared between amoxicillin and control treated pigs using a Wald parametric test in DESeq2 Bioconductor package in R using a false-discovery rate (FDR) threshold of 0.15 [25]. Only taxa present at $\geq$ 0.1% of all 16S rRNA sequences in either amoxicillin or control group were considered.

**mRNA-seq analysis.** Data analysis was performed by an individual blinded to *in vivo* sample identity (MC). Sequencing data, up to the raw-counts stage, was analyzed using the GenPipes pipeline [26]. FastQ files containing raw reads were trimmed using Trimmomatic and filtered for quality [27]. Filtered, high-quality reads were aligned to the *Sus scrofa* genome (Ensembl Sscrofa 11.1) using the STAR aligner 2-passes mode [28]. Samtools (v1.1[29]) was used to sort the BAM alignment files and to convert them into SAM format. These files were used as input for HTSeq-count, to generate a matrix of the number of reads per gene [30]. This raw-counts matrix was imported into the iDEP (v0.82) framework for downstream analysis [31], which included sample filtering (at least 0.05 read counts per million, CPM, in at least 4/40 samples), count normalization (variance stabilizing transformation, VST, as per the DESeq2 package), exploratory data analysis (Principal component analysis, PCA, and hierarchical clustering using correlation coefficient distances and average-linkage), differential expression (DESeq2 package, FDR <0.1 based on the Benjamini-Hochberg method, fold-change > 2 [25]). Pathway analysis was performed using GSEA and Ingenuity Pathway Analysis (v46901286, IPA, Qiagen, Mississauga, ON) on expression log values generated by DESeq2 [32]. Individual pathways were scored based on their predicted activation (positive Z score) or inhibition (negative Z score), and only pathways with an absolute score > 0.5 were reported.

## Results

### Histopathology

A total of 120 explant sections (10 pigs x 2 groups (amoxicillin, placebo) x 3 treatments (PBS, LPS, *Salmonella*) x 2 time points (0, 30 minutes) were scored for cellular death of the

superficial epithelium layer. Extensive mats of bacilli were found in *Salmonella*-exposed explants interacting with and invading the superficial epithelial layer (Fig 1A). Necrosis scores in *Salmonella* and LPS-exposed samples were significantly higher than the negative control PBS group ($P < 0.05$, Fig 1B), regardless of the *in vivo* antibiotic treatment. *Salmonella* and LPS groups did not statistically differ in necrosis scores. While high scores for epithelial necrosis were observed in these two groups, stroma cells were normal in appearance (Fig 1A).

## Bacterial communities

Rectal swabs collected from pigs at 7 and 21 days of age (during and after amoxicillin treatment), before washing and from explants immediately before placement in the incubator were analyzed (n = 20/sampling event) by high-throughput sequencing following amplification of the 16S rRNA gene. After quality-control steps, an average of 20797 reads (ranging from 2716 to 62790, totalling 1871782) were used for downstream analysis.

The rectal microbial community structure differed between amoxicillin and control pigs on day 7 (Bray-Curtis dissimilarity, $P = 0.014$, Fig 2A). A significant reduction in species richness and diversity was also observed (Chao1: $P = 0.033$, Shannon Index: $P = 0.024$, Fig 2B). A larger proportion of Actinobacteria and Fusobacteria was associated with the control group ($P < 0.05$, Fig 2C and 2D), while amoxicillin-treated pigs had a larger proportion of Proteobacteria ($P = 0.05$, Fig 2D). Based on the taxa enrichment analysis, the genera *Enterococcus*, *Blautia*, *Coriobacteriaceae*, *and Lactobacillus* were significantly enriched in piglets treated with amoxicillin ($P < 0.05$, Fig 2D), and also was the family Enterobacteriaceae ($P \leq 0.10$, Fig 2D); whereas the genus *Veillonella* and the family Mogibacteriaceae tended to be enriched in the controls ($P \leq 0.10$ and $P < 0.05$, respectively, Fig 2D).

On day 21, the gut-associated microbial structure of amoxicillin and control piglets differed significantly (Bray-Curtis dissimilarity, $P = 0.01$, Fig 3A), and the phyla Verrucomicrobia and Synergistetes were enriched in amoxicillin treated piglets ($P < 0.05$, $P \leq 0.10$ respectively, Fig 3C). However, no differences in species richness and diversity were observed between amoxicillin and control piglets (Chao1: $P = 0.453$, Shannon Index: $P = 0.562$, Fig 3B). The genera *Eubacterium*, *Parabacteroides*, *Akkermansia Christensenella*, and family Rikenellaceae were enriched in amoxicillin treated piglets ($P < 0.05$, Fig 3D). The genera *Corynebacterium*, *ph2*, *Methanobrevibacter* and *Pyramidobacter* tended to be enriched in amoxicillin treated piglets ($P \leq 0.10$). In the control piglets only *Odoribacter* was significantly enriched ($P < 0.05$) and *Prevotella* and *Dialister* tended to be enriched on day 21 ($P \leq 0.10$).

In the post-washing mucosal swabs, 4 samples from amoxicillin treated pigs had to be removed from the study due to insufficient DNA yield after extraction. Post-washing, the mucosa-associated microbiota structure of amoxicillin and control pigs differed significantly (Bray-Curtis dissimilarity, $P = 0.01$, Fig 4A). However, no differences were found regarding alpha-diversity metrics (Chao1: $P = 0.543$, Shannon Index: $P = 0.168$, Fig 4B). Furthermore, no differences were found between post-mortem and post-washing samples (S1 Fig).

## Transcriptome analysis

A total of 25879 genes were detected from the control and *Salmonella* exposed explants (LPS positive control group not analyzed). After filtering out infrequently-expressed genes (<5000 reads/million), 19268 genes were retained, from which 15921 were converted to Ensembl gene IDs. The other 3347 genes were kept in the data using original IDs, but were not included in the pathway analysis. Principal component analysis revealed clusters of samples grouped by *in vivo* antibiotic treatment (amoxicillin or PBS), and *in vitro* challenge (*Salmonella* or control, Fig 5). Individual gene expression analysis, comparing *in vivo* groups (amoxicillin versus

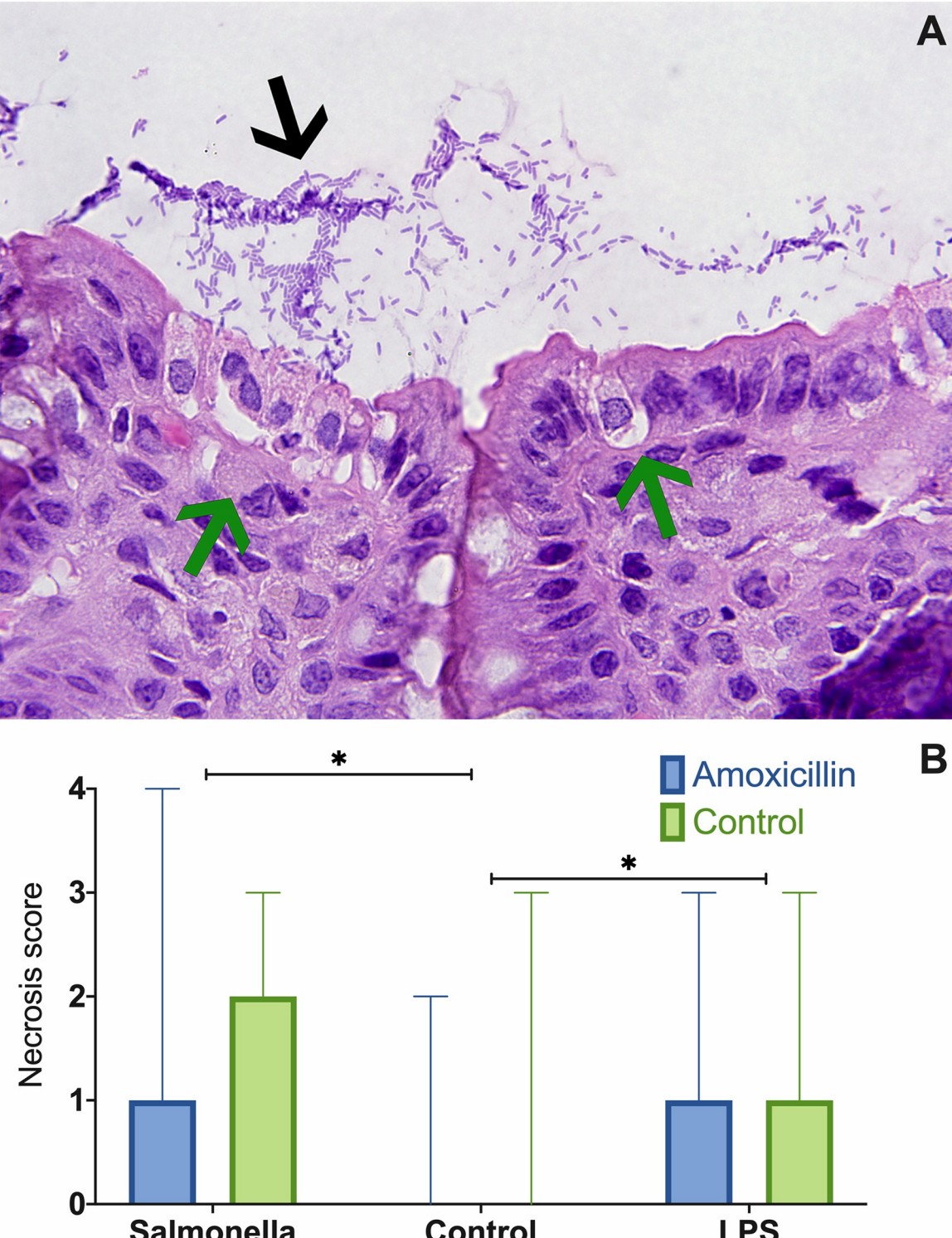

**Fig 1. H&E stained section of explant inoculated with *Salmonella enterica* Typhimurium, from a pig not treated with amoxicillin.**
Loss of superficial epithelial layer cells (green arrows) and a mat of bacilli (black arrow) is observed (**A**). Bar chart (median and range) based on necrosis scores from H&E stained explant sections. Stars denotes differences between inoculum groups ($P < 0.05$, **B**).

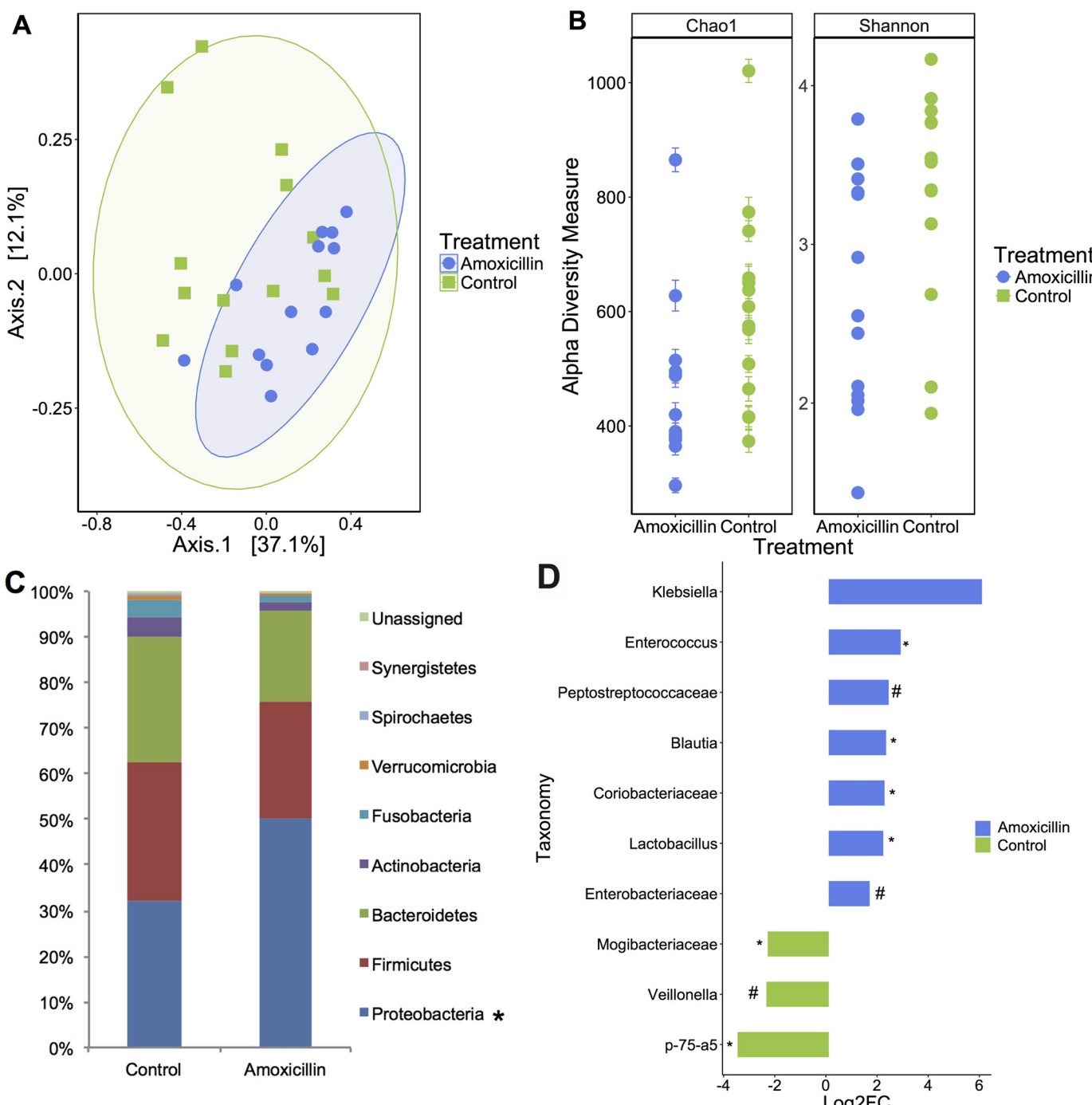

**Fig 2.** Amoxicillin treatment influenced gut-associated microbial composition, as displayed by distinct clustering in the principal coordinates plot of Bray Curtis dissimilarity on day 7 (**A**). Each dot represents fecal microbiota of one piglet (Anosim *P* = 0.014). Alpha diversity was significantly reduced in piglets treated with amoxicillin on day 7 as indicated by Chao1 and Shannon index (*P* = 0.033 and *P* = 0.024, respectively, **B**). The stacked bar chart displays predominating phyla, calculated as a percentage of total 16S rRNA reads (**C**). Taxonomic differences between amoxicillin and placebo-treated piglets were identified by DeSEq2. Taxa enriched in amoxicillin are shown in blue; taxa enriched in placebo piglets are shown in green (**D**). (*P* < 0.05; #*P* ≤ 0.10).

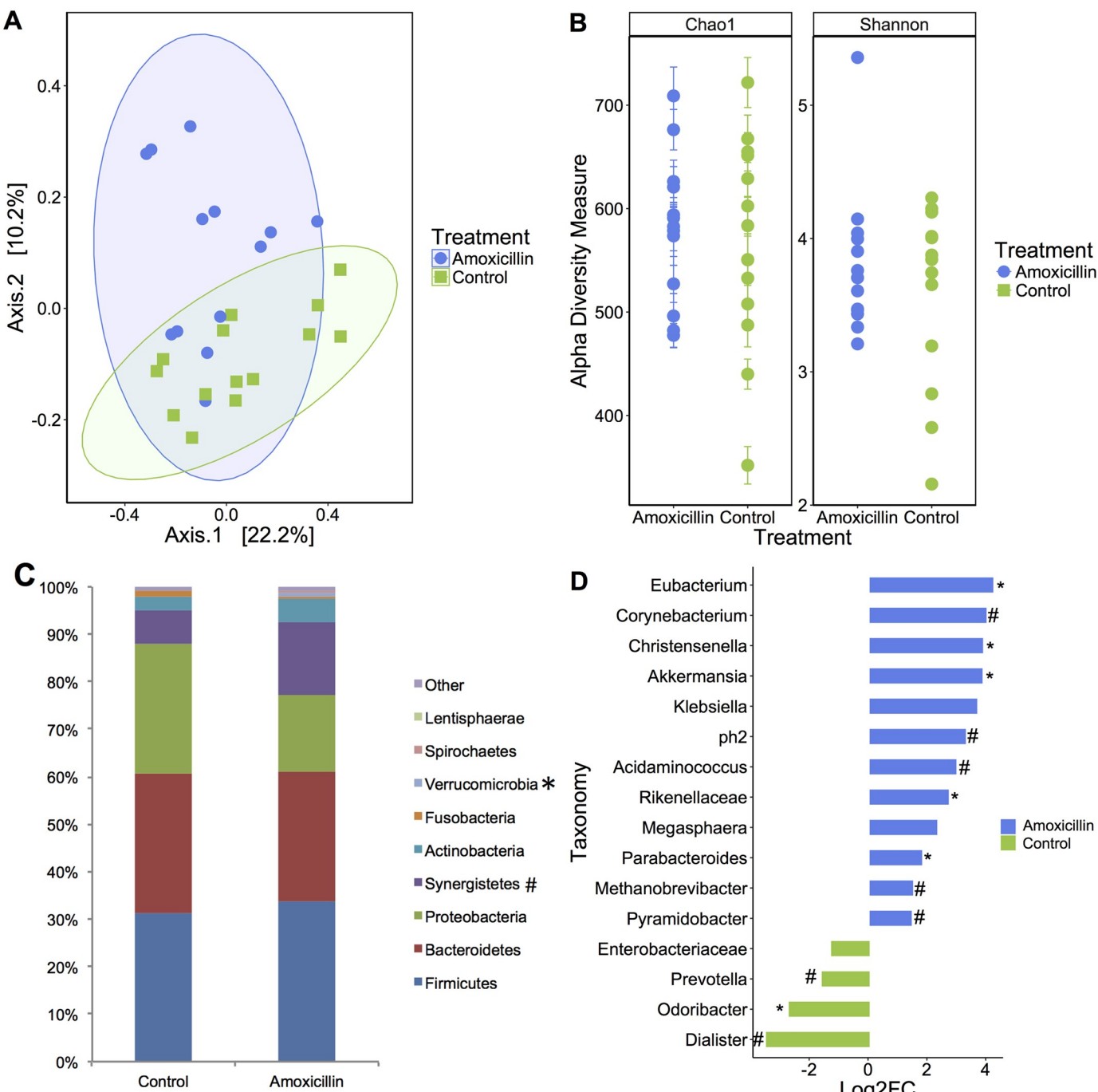

**Fig 3.** Amoxicillin treatment influenced gut-associated microbial composition, as displayed by distinct clustering in the principal coordinates plot of Bray Curtis dissimilarity on day 21 (**A**). Each dot represents fecal microbiota of one piglet (Anosim $P = 0.01$). Alpha diversity was no longer distinct in piglets treated with amoxicillin on day 21 as indicated by Chao1 and Shannon index ($P = 0.453$ and $P = 0.562$, respectively, **B**). Stacked bar chart displays predominating phyla, calculated as a percentage of total 16S rRNA reads (**C**). Taxonomic differences between amoxicillin and placebo-treated piglets were identified by DeSEq2. Taxa enriched in amoxicillin are shown in blue; taxa enriched in placebo piglets are shown in green (**D**). (*$P < 0.05$; #$P \leq 0.10$).

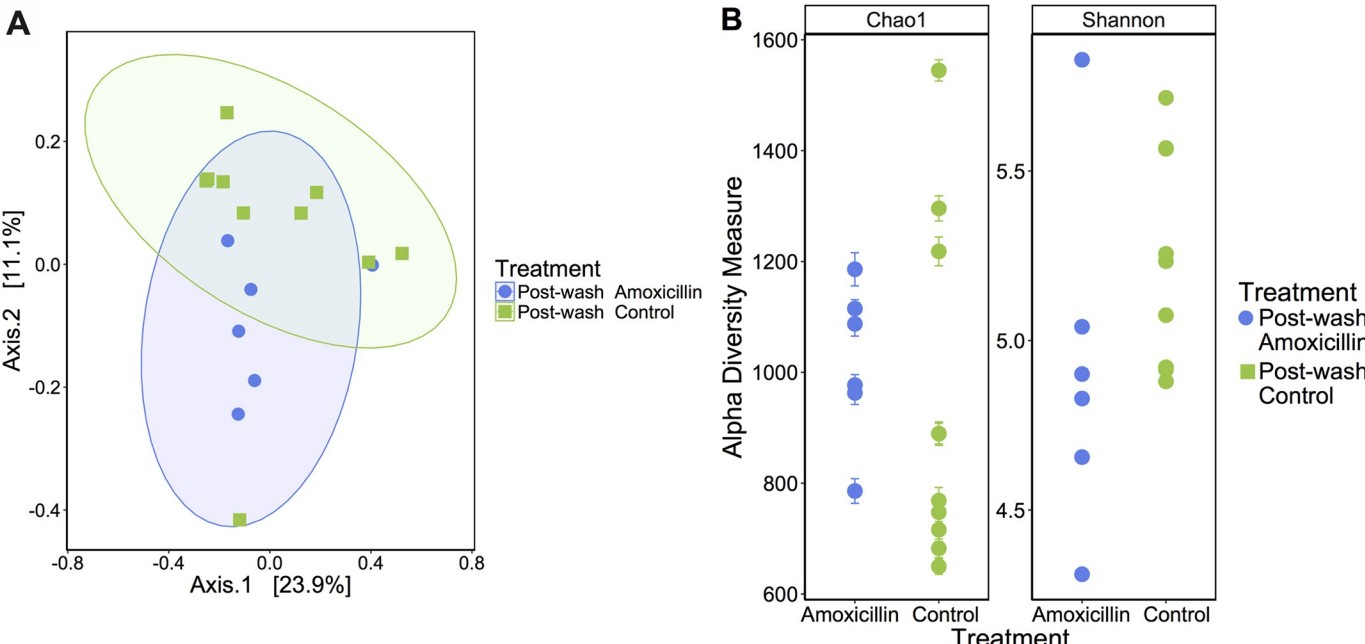

**Fig 4.** Amoxicillin treatment influenced gut-associated microbial composition, as displayed by distinct clustering in the principal coordinates plot of Bray Curtis dissimilarity on day 21 post tissue washing (**A**). Each dot represents fecal microbiota of one piglet (Anosim $P = 0.01$). Alpha diversity was not significantly reduced in piglets treated with amoxicillin on day 21 post tissue washing as indicated by Chao1 and Shannon index ($P = 0.543$ and $P = 0.168$, respectively) (B).

placebo pigs in PBS explants only) identified 2 genes differentially expressed between amoxicillin and placebo-treated pigs ($P < 0.05$). Specifically, BPIFB2 (1.32-fold change, BPI fold-containing family B member 2) was up-regulated and PIMREG (-1.09-fold change, PICALM interacting mitotic regulator) was down-regulated in explants from amoxicillin-treated pigs. Regardless of *in vivo* antibiotic treatments, two genes were down-regulated in the *Salmonella* exposed explants compared to PBS control explants (1.08-fold change, BPIFB2, and 1.72-fold change, MARCKSL1. S2 Table).

To further understand how effects of amoxicillin treatment *in vivo* and Salmonella exposure *in vitro*, functional transcriptome analysis was performed. Gene-set enrichment analysis (GSEA) of control explants from amoxicillin and placebo-treated pigs demonstrated that amoxicillin led to the activation of biological processes associated with the immune response, including a response to intracellular pathogens and antigen presentation, and the inhibition of biological processes associated with muscular activity (Table 1). Next, we investigated individual molecular pathways affected by each treatment (summarized in Fig 6, only significant pathways with a Z-score > |0.5| included). Various mechanisms of tissue repair and cellular replenishment were activated in amoxicillin-exposed explants compared to the placebo group: signaling by Rho Family GTPases (Z-score = 3.0), integrin signaling (2.8) and estrogen-mediated S-phase entry (2.0, Fig 6A). The following pro-inflammatory pathways had the highest scores in the *Salmonella* versus PBS-exposed contrast, regardless of *in vivo* antibiotic group (Fig 6B): neuropathic pain signalling (4.6), dopamine-DARPP32 feedback in cAMP signaling (3.4), fMLP signalling in neutrophils (3.1), and P2Y receptor signaling (2.9) as well as the GNRH signaling pathway (3.0). The anti-inflammatory endocannabinoid pathway was significantly inhibited in the *Salmonella* group (-2.6). When comparing *Salmonella* and PBS explant

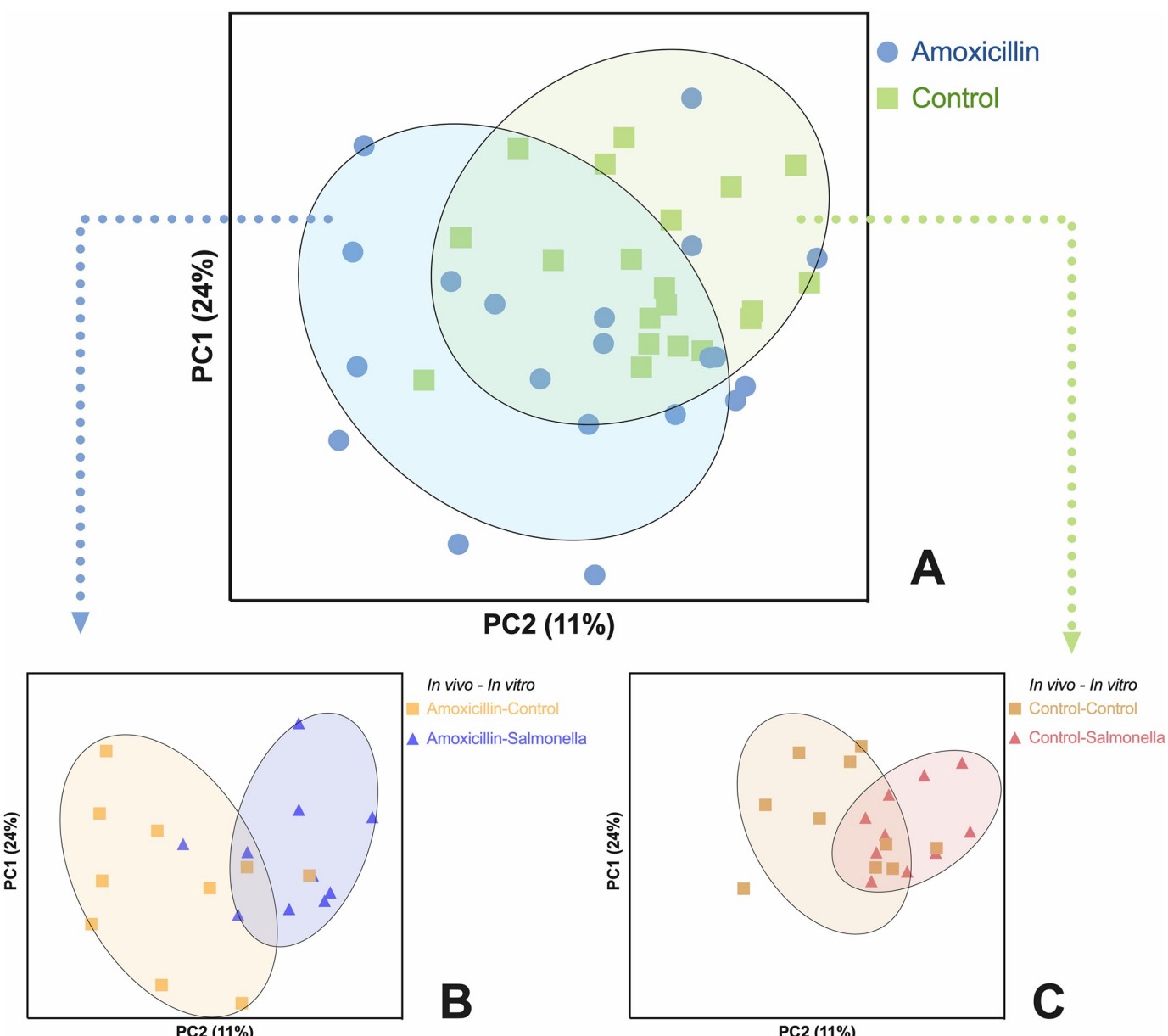

**Fig 5. Principal component analysis based on whole transcriptomes.** All samples included, both *in vivo* and *in vitro* groups (**A**); *Salmonella* versus PBS control explants in amoxicillin-treated pigs (**B**); *Salmonella* versus PBS control explants in placebo-treated pigs (**C**).

groups from amoxicillin-treated pigs, a pro-inflammatory process was observed in *Salmonella*-challenged explants. Highest scores were associated with B cell receptor signalling (3.4), the activation of IL-2 expression in activated and anergic T lymphocytes (2.5), PPAR (peroxisome proliferator-activated receptors) signalling (-2.8) and B cell activating factor signalling (2.0) pathways (Fig 6C). Finally, *Salmonella*-infected explants from amoxicillin compared to placebo-treated pigs had the following pathways as the highest scoring: NER (nucleotide excision repair, 4.8), pyrimidine de novo and salvage pathways (2.6) and activation of nitric oxide (NO) production pathway (iNOS, 2.4, Fig 6D).

**Table 1. Top 30 significantly affected biological process identified by geneset enrichment analysis (GSEA) in control explants from amoxicillin and placebo-treated pigs.**

| Direction | GO terms | NES* | Genes | adj.Pval |
|---|---|---|---|---|
| **Down** | Muscle contraction | -1.9689 | 196 | 0.0016 |
| | Myofibril assembly | -1.9651 | 52 | 0.0022 |
| | Striated muscle contraction | -1.9622 | 107 | 0.0016 |
| | Cardiac muscle cell development | -1.9392 | 42 | 0.0022 |
| | Cardiac cell development | -1.933 | 44 | 0.0029 |
| | Muscle system process | -1.905 | 245 | 0.0016 |
| | Striated muscle cell development | -1.9043 | 112 | 0.0016 |
| | Multicellular organismal movement | -1.8963 | 41 | 0.0022 |
| | Musculoskeletal movement | -1.8963 | 41 | 0.0022 |
| | Skeletal muscle contraction | -1.8955 | 28 | 0.0030 |
| | Heart contraction | -1.8903 | 144 | 0.0016 |
| | Regulation of system process | -1.8853 | 299 | 0.0016 |
| | Collagen fibril organization | -1.8796 | 48 | 0.0035 |
| | Adenylate cyclase-inhibiting G protein-coupled receptor signaling pathway | -1.877 | 46 | 0.0035 |
| | Sarcomere organization | -1.8675 | 35 | 0.0057 |
| | Trophectodermal cell differentiation | -1.8645 | 15 | 0.0051 |
| | Regulation of neurological system process | -1.8621 | 50 | 0.0035 |
| **Up** | Regulation of response to biotic stimulus | 2.0829 | 94 | 0.0023 |
| | Antimicrobial humoral immune response mediated by antimicrobial peptide | 2.0245 | 32 | 0.0022 |
| | Antigen processing and presentation of peptide antigen via MHC class I | 2.0011 | 17 | 0.0039 |
| | Positive regulation of response to biotic stimulus | 1.9796 | 36 | 0.0056 |
| | Antigen processing and presentation of peptide antigen | 1.9794 | 27 | 0.0069 |
| | Defense response to Gram-negative bacterium | 1.9626 | 34 | 0.0050 |
| | Regulation of defense response to virus | 1.9573 | 51 | 0.0051 |
| | Organ or tissue specific immune response | 1.9479 | 18 | 0.0081 |
| | Mucosal immune response | 1.8768 | 16 | 0.020 |
| | Regulation of defense response to virus by host | 1.8762 | 26 | 0.020 |
| | Cellular response to virus | 1.8703 | 40 | 0.014 |
| | Antigen processing and presentation | 1.8579 | 59 | 0.068 |
| | Antimicrobial humoral response | 1.8579 | 45 | 0.014 |

*- Normalized enrichment score.

## Discussion

Salmonellosis is a concern to public health and the swine industry. it can cause diarrhea and septicaemia in pigs, and the organism may contaminate pork products during the slaughter process, potentially infecting humans. *Salmonella* Typhimurium is the serovar most frequently isolated from swine in North America, and it is commonly associated with diarrhea and enterocolitis in pigs [33]. Reducing clinical disease can help improve animal productivity and decrease the incidence of pork-associated Salmonellosis cases in humans. Exploiting the gut bacterial community as a tool to reduce clinical disease severity, pathogen shedding and, potentially, pork contamination, can be beneficial as a mass prevention strategy to improve animal welfare and reduce the need of antibiotics for high-quality protein production. The overarching goal of the work presented here was to determine if modulation of the colonic microbiome *in vivo*, through oral amoxicillin treatment, affects the host response to *Salmonella* Typhimurium infection *in vitro* using an organ explant model.

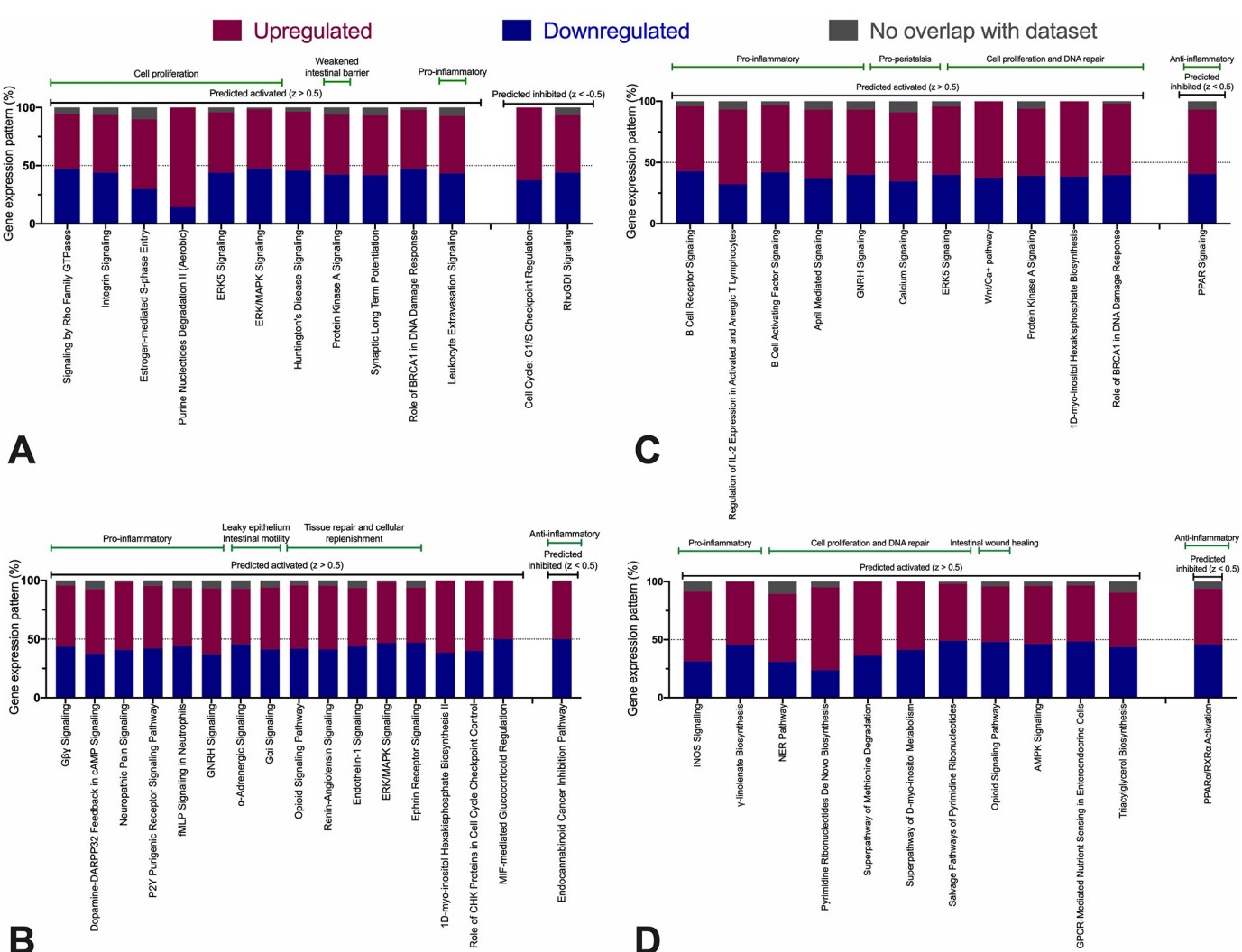

**Fig 6. Percentage of upregulated and downregulated genes (including genes not present in the dataset) associated with pathways that significantly differed between groups.** Predicted activated or inhibited pathways according to z-test for each pathway are also shown. Gene expression pattern should be interpreted in relation to the challenge group (amoxicillin or *Salmonella*). Green bars exemplify the biological significance of each pathway. Black bars denote pathway activation or inhibition. *In vivo* amoxicillin versus placebo treated pigs (**A**); *Salmonella* versus PBS control explants, regardless of *in vivo* treatment (**B**); *Salmonella* versus PBS control explants from amoxicillin-treated pigs only (**C**); *Salmonella* explants only from amoxicillin versus placebo treated pigs (**D**).

We applied high-throughput sequencing to profile both the rectal bacterial communities of suckling pigs treated with amoxicillin, and gene expression in colon explants in response to *Salmonella* infection *in vitro*. At 7 days of age amoxicillin-treated pigs were enriched for Enterobacteriaceae, which has been previously observed in pigs [14]. At 21 days of age, our results indicated increasing amounts of the Verrucomicrobia and Synergistetes phyla in rectal swab samples of amoxicillin-treated pigs. This observation was associated with a lower number of necrotic enterocytes detected in explants following *Salmonella* exposure, as well as the activation of cell proliferation, DNA repair and intestinal wound healing (opioid signalling) mechanisms [34]. It has been previously reported that the gut microbiome can alter the host gene expression pattern [35–38] and it is important to highlight that the *in vitro* microbiota in our explant model resembled the *in vivo* microbiota from donors (S1 Fig). Previous studies have demonstrated that the microbiome can be leveraged to protect the host from pathogen

colonization or overgrowth of indigenous pathobionts, a concept thoroughly reviewed by Kamada *et al* [39]. We observed a reduced number of necrotic mucosal epithelial cells in explants from amoxicillin-treated pigs (compared to placebo pigs) following *Salmonella* exposure, linked to activation of the iNOS-dependent nitric oxide production cascade. We acknowledge the high variation in necrosis scores, which are postulated to be linked to bacteriome diversity. It has been described that production of oxygen reactive species by iNOS leads to increased oxidative stress, and one of the consequences was the facilitated clearance of intestinal pathogens in a mouse model of infectious colitis [40, 41]. Although *Salmonella* has evolved to resist host-produced reactive oxygen species, we hypothesize that the *in vivo* antibiotic treatment led to a more intense host response, rendering the bacterium defenses insufficient, leading to a reduced number of necrotic cells [42]. In association with the activation of the iNOS pathway, we also identified the activation of multiple cell proliferation pathways in explants of amoxicillin-treated pigs. Epithelial cell proliferation has been described as a resilience marker in mice resistant to *Citrobacter rodentium* colitis [41, 43]. Other factors contributing to this putative resilience to *S*. Typhimurium cell damage are not clear, but the increased proportion of Synergistetes in amoxicillin-treated pigs may be associated with this event. Higher Synergistetes ratios in the human fecal microbiome was positively correlated with increased total and anti-phosphorylcholine (PC) IgM [44]. This particular type of IgM was described as a key factor capable to potentialize the phagocytosis of apoptotic cells and to inhibit proinflammatory pathways in autoimmunity and atherosclerosis [45–47]. These are potential mechanisms that may have contributed to the reduced number of necrotic cells observed in explants from amoxicillin-treated pigs following challenge with *S*. Typhimurium. Noteworthy, a previous study using mice also attempted to show the beneficial effects of antibiotic-based microbiome modulation. The authors reported that the development lesions following *Campylobacter jejuni* or *Acinetobacter baumannii* inoculation was not affected by antibiotic treatment, despite inducing dysbiosis [48].

Although our investigation did not reveal a large number of individual genes differentially expressed between groups, gene set analysis did identify uniquely affected pathways. Having only a few differentially expressed genes was unexpected, but it can be explained by the short duration of the *in vitro* challenge. This timeframe was utilized based on pilot studies revealing significant intestinal epithelial necrosis after 30 minutes of pathogen exposure in our IVOC model. In light of this, we applied GSEA as it has been shown to be a more sensitive approach to evaluate transcriptomic changes than individual gene expression fluctuations [32]. In any living organism, multiple genes are linked to a single biological pathway, and this additive effect in expression within pathways is associated with phenotypical differences. GSEA takes advantage of gene groups defined *a priori* to identify changes in expression that, together, affect an entire pathway. Using this approach, we observed that the transcriptome of control explants differed between amoxicillin-treated and control pigs. Overall, biological processes associated with muscle cell metabolism were down-regulated, while processes related to the immune-response were up-regulated. It is known that intestinal bacterial communities are part of the intestinal homeostasis, including immunomodulation. Therefore, disruption of these communities by antibiotics appears to exert an indirect effect on intestinal immunological function. Several studies have investigated the effects of antibiotics on the pig intestinal transcriptome, showing that antibiotic treatment in healthy animals can decrease the production of IFN-γ and Th17-producing lymphocytes, and reduce the expression of pro-IL-18 and pro-IL-1β and change TLR expression patterns [49–51]. These effects, however, should be extrapolated cautiously because changes in the microbiota are suggested to be antibiotic-specific. The β-lactam ampicillin leads to decreased bacterial diversity and increased prevalence of *Enterobacter* spp [52]. Streptomycin, a aminoglycoside, also leads to decreased diversity while

enriching for Ruminococcaceae and Bacteroidaceae [53]. Tigecycline, a tetracycline, induces a reduction in the abundance of Bacteroidetes and increases the abundance of Proteobacteria [54]. Oral administration of vancomycin, a glycopeptide, decreases bacterial diversity, enriches for Proteobacteria and Tenericutes while depleting Bacteroidetes and Firmicutes and affecting intestinal carbohydrate and lipid metabolism [55, 56] Similarly to what we observed, another study also reported the positive regulation of gene ontology (GO) terms related to immune effector processes in healthy pigs receiving long-term in-feed antibiotics [57]. In contrast, *S.* Typhimurium has been suggested to thrive in the pro-inflammatory environment created by itself once it invades the mucosa and infect macrophages [13]. Following mucosal invasion, *S.* Typhimurium induces interleukin (IL)-22 and IL-17 transcription to elicit an innate immune response against luminal bacteria (mediated through bacteriocins), while increasing mucus secretion. This changes the surrounding microbiota composition, while increasing *S.* Typhimurium source of energy carbohydrates, thus conferring a nutritional advantage. In corroboration with this previously described mechanism, we identified the inhibition of PPAR signalling in *Salmonella*-challenged explants. The PPAR pathway has been linked to intestinal microbiota modulation through regulation of IL-22 and antimicrobial peptides secretion [58].

Taken together, the data presented here suggests that early life enteral administration of amoxicillin modulated the intestinal microbiota of piglets, enriching these bacterial populations for Synergistetes and other taxa (Fig 3). This particular dysbiosis induced the activation of pathways associated with the immune system that may have primed the intestinal mucosa to respond quickly to pathogens, such as *S.* Typhimurium (Table 1). Specific pathways activated were related to B cell receptor (BCR) activation, IL-2 and NO production (Fig 6). In parallel, we also detected the activation of pathways related to intestinal wound healing and cell proliferation, which suggest an increased capacity of tissue regeneration (Fig 6). The concept of microbiome modulation suggested above may become applicable to livestock if a suitable non-antibiotic alternative to amoxicillin is developed. Prebiotics have previously been used to enrich for beneficial bacteria (*Bifidobacterium* spp.), leading to significant clinical improvement in humans affected with chronic colitis [59, 60]. Here, we suggest that future investigations may revolve around the identification of selective fermented feed ingredient that will result in the microbiota changes similar to what was observed in this study. As our understanding of the host-microbiota-pathogen axis increases, there is growing interest in how prebiotics can be applied to exploit these interactions to benefit the host [61]. Overcoming this challenge will allow for the large-scale microbiome modulation without the risk of inducing antimicrobial resistance.

In conclusion, *in vivo* treatment of pigs with amoxicillin in early life altered the gut microbiota, affected the host gene expression pattern which was linked to a decreased number of necrotic intestinal epithelial cells following *in vitro* challenge with *S.* Typhimurium using an organ culture model. Further investigations are warranted to verify these observations *in vivo*. The authors also recognize that, moving forward, studies focused on the identification of key players (at any taxonomic level) of the immunostimulatory effect observed here will be required. Finally, the development of strategies to modulate the indigenous intestinal microbiota without the use of antibiotics are necessary to make such approach applicable to livestock species. In face of the rise in antibiotic resistant bacteria, such approach cant greatly contribute to the efforts towards judicious use of antibiotics for food production.

## Supporting information

**S1 Fig.** No distinctions were observed in microbial composition pre and post-washing of explants within each treatment as displayed by the indistinct clustering between pre and post

wash within treatment groups (Amoxicillin and Control) in the principal coordinates plot of Bray Curtis dissimilarity (**A**). Alpha diversity did not differ with tissue washing within treatment groups (Amoxicillin and Control) as measured by Chao1 and Shannon Index (**B**). (TIFF)

**S1 Table. Summary of piglet data (weight, amox dose and age at euthanasia).**
(XLSX)

**S2 Table. Log fold change and adjusted P-values for all genes.**
(XLSX)

## Author Contributions

**Conceptualization:** Matheus O. Costa, Benjamin Willing, John C. S. Harding.

**Data curation:** Matheus O. Costa.

**Formal analysis:** Matheus O. Costa, Janelle Fouhse.

**Funding acquisition:** John C. S. Harding.

**Investigation:** Matheus O. Costa, Ana Paula P. Silva.

**Methodology:** Matheus O. Costa.

**Project administration:** Matheus O. Costa, John C. S. Harding.

**Resources:** Matheus O. Costa.

**Software:** Matheus O. Costa.

**Supervision:** Matheus O. Costa.

**Validation:** Matheus O. Costa.

**Visualization:** Matheus O. Costa.

**Writing – original draft:** Matheus O. Costa.

**Writing – review & editing:** Matheus O. Costa, Benjamin Willing, John C. S. Harding.

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
