## [Decision Letter · Decision Letter 0]

9 Dec 2019

PONE-D-19-19473

Putting the microbiota to work: swine intestinal microbiota modulation by antibiotic treatment is linked to the up-regulation of immune-associated pathways using an in vitro Salmonella Typhimurium challenge model

PLOS ONE

Dear Dr. de Oliveira Costa,

Thank you for submitting your manuscript to PLOS ONE. After careful consideration, we feel that it has merit but does not fully meet PLOS ONE’s publication criteria as it currently stands. Therefore, we invite you to submit a revised version of the manuscript that addresses the points raised during the review process.

We would appreciate receiving your revised manuscript by Jan 23 2020 11:59PM. To enhance the reproducibility of your results, we recommend that if applicable you deposit your laboratory protocols in protocols.io, where a protocol can be assigned its own identifier (DOI) such that it can be cited independently in the future. For instructions see: http://journals.plos.org/plosone/s/submission-guidelines#loc-laboratory-protocols

We look forward to receiving your revised manuscript.

Kind regards,

Praveen Rishi, Ph.D., FAMI, FABMS

Academic Editor

PLOS ONE

Journal Requirements:

2.  Please amend either the title on the online submission form (via Edit Submission) or the title in the manuscript so that they are identical.

3. Please upload a copy of Figure 7, to which you refer in your text on page xx. If the figure is no longer to be included as part of the submission please remove all reference to it within the text.

4. Please ensure that you refer to Figure 6 in your text as, if accepted, production will need this reference to link the reader to the figure.

Reviewers' comments:

Reviewer's Responses to Questions

**Comments to the Author**

1. Is the manuscript technically sound, and do the data support the conclusions?

Reviewer #1: No

Reviewer #2: Yes

2. Has the statistical analysis been performed appropriately and rigorously? 

Reviewer #1: No

Reviewer #2: Yes

3. Have the authors made all data underlying the findings in their manuscript fully available?

Reviewer #1: No

Reviewer #2: Yes

4. Is the manuscript presented in an intelligible fashion and written in standard English?

Reviewer #1: No

Reviewer #2: Yes

5. Review Comments to the Author

Reviewer #1: Manuscript entitled “Putting the microbiota to work-------------- Salmonella Typhimurium challenge” submitted by Costa et al. deals with modulation of intestinal microbiome in piglets with antibiotic for 14 days and is of the opinion that antibiotic treatment in early life alters the gut microbiome resulting into decreased necrotic intestinal lesions following S. Typhimurium challenge. However, the proposed mechanism of gut modulation by antibiotic treatment in early phase of life cannot be correlated with the present study and their basis of discussion is purely hypothetical and is not supported with facts and findings.

Specific comments

1) Title needs to be changed

2) Abstract is not highlighting the important findings written properly.

3) Introduction is not written properly

4) No of replicates should be mentioned in material and methods

5) Transcriptomics data visualization is confusing

6) Data needs to be monitored statistically and carefully as in fig 1 the standard deviations too high

7) Figure no is not matching with text

8) Differential gene expression analysis are not appropriate in both the conditions.

9) Validation of RNA sequencing of data should have been supported by qPCR

10) Table 1 is very confusing. It is surprising to see the results of table 1 where authors have indicated down and up regulation of various biological processes. On the basis of which the authors have concluded the up regulation of immune response genes as the protective mechanism for intestinal pathogens. However, they have ignored the genes responsible for the basic physiology of living beings such as muscle contraction, heart contraction etc. that too are very significant in day to day life of living beings.

11) Manuscript is not discussed properly as the proposed modulatory mechanism is not clear.

12) Grammatically paper needs to be edited as English is very poor.

Overall, the manuscript is not acceptable.

Reviewer #2: The present manuscript describes the effect of amoxicillin treatment on the normal flora, immune-associated pathways and subsequent exposure to Salmonella Typhimurium in a piglet model. My specific concerns about the work are described below:

- The study has been executed well. However, the manuscript will benefit if the authors can improvise on the basis of this study, and the final conclusions that should be interpreted. Is the manuscript communicating amoxicillin exposure to be a viable option for protecting against subsequent Salmonella infection? If yes, that may not be a feasible alternative, considering the current trends in antimicrobial resistance resulting from livestock/poultry exposure. If no (as the authors themselves conclude in Lines 407-408), then how should indigenous microflora be modulated to mimic the antibiotic and the results obtained? These points may be discussed and added to the manuscript.

- Lines 118-119: the chambers are written/named as anaerobic, but the oxygen concentration is 95%. The statement may be reframed.

- The authors may clarify if the no. of explants per pig were 20 (Line 33) or 18 (Line 109).

- Lines 120, 121: the sentence “After 0 and 30 min of co-culture, explants were fixed with formalin and RNAlater” needs to be reframed for language.

- Line 139: “plate reading guidelines”, guidelines may not be the right word for just stating that the interpretations were being done by checking the colony color on the basis of a principle (Lines 131-132).

-Lines 261-262: some of the interpretation/results in text are not matching with the figure. At day 7, the Enterobacteriaceae appears enriched in amox-treated pigs than controls.

Line 389” “should extrapolated cautiously because changes in the microbiota are antibiotic-specific.” The manuscript will benefit if this statement is cited with appropriate examples, of antibiotic classes and the resulting changes.

6. PLOS authors have the option to publish the peer review history of their article (what does this mean?). If published, this will include your full peer review and any attached files.

Reviewer #1: No

Reviewer #2: No

---

## [Author Response · Author response to Decision Letter 0]

24 Jan 2020

Reviewer #1: Manuscript entitled “Putting the microbiota to work-------------- Salmonella Typhimurium challenge” submitted by Costa et al. deals with modulation of intestinal microbiome in piglets with antibiotic for 14 days and is of the opinion that antibiotic treatment in early life alters the gut microbiome resulting into decreased necrotic intestinal lesions following S. Typhimurium challenge. However, the proposed mechanism of gut modulation by antibiotic treatment in early phase of life cannot be correlated with the present study

-We have revised the discussion to address the lack of correlation suggested by the reviewer. Please see lines 497-504. We have also performed additional experiments (that will be submitted for publication soon) and confirmed that amoxicillin treatment early in life does lead to decreased Salmonella shedding in nursery-aged pigs.

and their basis of discussion is purely hypothetical and is not supported with facts and findings.

-The discussion is based on the findings presented in the results section. The goal of this study was to test the hypothesis that amoxicillin treatment in vivo can reduce epithelial death in vitro, following salmonella inoculation. We believe we have tested that hypothesis, and proven it to be true. The dissection of which specific mechanisms were responsible for such observation are discussed and suggested, but go confirmation of the relevance of such mechanisms are beyond the scope of the study. The authors believe that several studies will be needed in order to clarify all the mechanisms responsible for such observation, and that the work described here is merely a first step, a proof-of-concept study, that evidences the application of the theory proposed and provides initial data towards possible mechanisms.

Specific comments

1) Title needs to be changed

-We have addressed this concern and proposed a new title: 

2) Abstract is not highlighting the important findings written properly.

-Acknowledged. The abstract was extensively reviewed and improved.

3) Introduction is not written properly

-The introduction was re-written to address this concern.

4) No of replicates should be mentioned in material and methods

-Number of replicates was mentioned in the M&M section. In this new version of the manuscript, number of replicates are described in line 98 for the number of pigs used, line 131 for the total number of explants per pig, and lines 138-144 for the number of explants/group.

5) Transcriptomics data visualization is confusing

-While we have opted to not use usual means to show the activation/inhibition of pathways, we believe Figure 6 does communicate this. In addition, it shows the specific number of genes affected per pathway, and the overall biological process affected. The authors believe this figure to be informative and, perhaps because it has an unusual presentation, may require a bit more attention from the reader.

6) Data needs to be monitored statistically and carefully as in fig 1 the standard deviations too high

Thank you for the suggestion. We would like to point out to the reviewer that Figure 1 depicts median and range of the data, not standard deviation. This presentation was chosen since this is categorical data, not continuous.

7) Figure no is not matching with text

Thank you for pointing that out. We have addressed the mismatch between the text (Figure 7) and the legends/files (figure 6).

8) Differential gene expression analysis are not appropriate in both the conditions.

We respectfully disagree with the reviewer’s comment. We would like to stress that we used bioinformatic tools largely applied by the scientific community to analyze mRNA sequencing data. Cufflinks has over 7000 citations, and DESEq2 over 13000. Finally, GSEA has been cited by over 5000 times. However, we are open to any suggestions that may improve the analyses.

9) Validation of RNA sequencing of data should have been supported by qPCR

-We agree with the reviewer that the standard practice in RNASeq studies is to validate data using qPCR. Most studies find a wealth of DEG genes, and don’t usually apply GSEA. Validation of GSEA/IPA data is hardly feasible, given the number of genes included in each pathway (ranging from 20-1300). We also believe that the two analytical methods used, GSEA and IPA, provide further evidence of the pathway analyses and validate each other. Other previously published studies that have not used confirmatory qPCR, but only GSEA, include: https://doi.org/10.1016/j.ccell.2016.11.004 , https://doi.org/10.1186/1471-2229-11-87 , including one from PLOS One https://doi.org/10.1371/journal.pone.0138782 .

10) Table 1 is very confusing. It is surprising to see the results of table 1 where authors have indicated down and up regulation of various biological processes. On the basis of which the authors have concluded the up regulation of immune response genes as the protective mechanism for intestinal pathogens. 

-This was based on the combination of GSEA and IPA analyses. Both methods found immune-associated pathways to correlate with amoxicillin treatments.

However, they have ignored the genes responsible for the basic physiology of living beings such as muscle contraction, heart contraction etc. that too are very significant in day to day life of living beings.

-We agree with the reviewer that such pathways are also significant in the day-to-day life of living beings. However, all the analyses performed here include solely the mucosa and lamina propriae of the colon (excluding muscularis mucosa). The identification of such pathways by GSEA was not further corroborated by IPA, and are likely related to other pathways that share messenger mechanisms (such as cAMP, cGMP, PPAR, etc…). Hence why they were not further explored.

11) Manuscript is not discussed properly as the proposed modulatory mechanism is not clear.

Acknowledged. We have include a paragraph outlining the proposed mechanism. We have also modified the discussion for clarity.

12) Grammatically paper needs to be edited as English is very poor.

We have revised the manuscript to improve readability.

Reviewer #2: The present manuscript describes the effect of amoxicillin treatment on the normal flora, immune-associated pathways and subsequent exposure to Salmonella Typhimurium in a piglet model. My specific concerns about the work are described below:

- The study has been executed well. However, the manuscript will benefit if the authors can improvise on the basis of this study, and the final conclusions that should be interpreted. 

- We acknowledge that conclusions and interpretations of the findings from this study are somewhat polemic. We have included a paragraph summarizing our conclusions.

Is the manuscript communicating amoxicillin exposure to be a viable option for protecting against subsequent Salmonella infection? If yes, that may not be a feasible alternative, considering the current trends in antimicrobial resistance resulting from livestock/poultry exposure. If no (as the authors themselves conclude in Lines 407-408), then how should indigenous microflora be modulated to mimic the antibiotic and the results obtained? These points may be discussed and added to the manuscript.

-The study described here is merely a first step, a proof-of-concept study, that evidences the application of the theory proposed and provides initial data towards possible mechanisms. Nowhere in the text, or in any other media, we support the extensive use of antibiotics as a preventative method. In fact, as the reviewer suggested, the goal is to reduce the use of antibiotics by developing novel methods to modulate the microbiome. The development of such methods is not the goal of this study. However, we do provide here the basis and justification for such studies, the concept that microbiome modulation can affect infection outcomes. Now that the mechanisms are somewhat unveiled, we may be able to further develop other strategies to replicate these without the use of antibiotics.

- Lines 118-119: the chambers are written/named as anaerobic, but the oxygen concentration is 95%. The statement may be reframed.

- The chambers are sold as anaerobic modular chambers, we agree with the reviewer that in this case, they were not kept under anaerobic conditions. The text has been properly modified.

- The authors may clarify if the no. of explants per pig were 20 (Line 33) or 18 (Line 109).

- Thanks for pointing that out. We have corrected the text, n=20. 

- Lines 120, 121: the sentence “After 0 and 30 min of co-culture, explants were fixed with formalin and RNAlater” needs to be reframed for language.

- Acknowledged. The sentence has been re-written.

- Line 139: “plate reading guidelines”, guidelines may not be the right word for just stating that the interpretations were being done by checking the colony color on the basis of a principle (Lines 131-132).

-Acknowledged. The text was adapted to reflect this suggestion.

-Lines 261-262: some of the interpretation/results in text are not matching with the figure. At day 7, the Enterobacteriaceae appears enriched in amox-treated pigs than controls.

-We appreciate that the reviewer attentively read this manuscript. We have addressed this specific issue (lines 300-301) and reviewed the remaining of the results section for mistakes. 

-Line 389” “should extrapolated cautiously because changes in the microbiota are antibiotic-specific.” The manuscript will benefit if this statement is cited with appropriate examples, of antibiotic classes and the resulting changes.

-Acknowledged. We have adjusted the text and referenced a study showing that different antibiotics lead to different changes in microbial populations composition (line 459).

---

## [Decision Letter · Decision Letter 1]

28 Feb 2020

PONE-D-19-19473R1

Putting the microbiota to work: Epigenetic effects of early life antibiotic treatment are associated with immune-related pathways and reduced epithelial necrosis following Salmonella Typhimurium challenge in vitro.

PLOS ONE

Dear Costa

Thank you for submitting your manuscript to PLOS ONE. After careful consideration, we feel that it has merit but does not fully meet PLOS ONE’s publication criteria as it currently stands. Therefore, we invite you to submit a revised version of the manuscript that addresses the points raised during the review process.

We would appreciate receiving your revised manuscript by Apr 13 2020 11:59PM. To enhance the reproducibility of your results, we recommend that if applicable you deposit your laboratory protocols in protocols.io, where a protocol can be assigned its own identifier (DOI) such that it can be cited independently in the future. For instructions see: http://journals.plos.org/plosone/s/submission-guidelines#loc-laboratory-protocols

We look forward to receiving your revised manuscript.

Kind regards,

Praveen Rishi, Ph.D., FAMI, FABMS

Academic Editor

PLOS ONE

Reviewers' comments:

Reviewer's Responses to Questions

**Comments to the Author**

1. If the authors have adequately addressed your comments raised in a previous round of review and you feel that this manuscript is now acceptable for publication, you may indicate that here to bypass the “Comments to the Author” section, enter your conflict of interest statement in the “Confidential to Editor” section, and submit your "Accept" recommendation.

Reviewer #1: All comments have been addressed

Reviewer #2: (No Response)

2. Is the manuscript technically sound, and do the data support the conclusions?

Reviewer #1: Partly

Reviewer #2: Partly

3. Has the statistical analysis been performed appropriately and rigorously? 

Reviewer #1: Yes

Reviewer #2: Yes

4. Have the authors made all data underlying the findings in their manuscript fully available?

Reviewer #1: (No Response)

Reviewer #2: Yes

5. Is the manuscript presented in an intelligible fashion and written in standard English?

Reviewer #1: No

Reviewer #2: Yes

6. Review Comments to the Author

Reviewer #1: Authors have tried to modify the manuscript entitled “Putting the microbiota to work-------------- Salmonella Typhimurium challenge in vitro” as per reviewer’s suggestions but it is still very confusing and not an exciting research paper. However, the proposed mechanism of gut modulation by antibiotic treatment in early phase of life cannot be correlated with the present study and their basis of discussion is still purely hypothetical.

Author should clarify whether they have used prophylactic or metaphylactic studies. I still feel that the article is very confusing and is not acceptable for publication.

Reviewer #2: In the revised version of the manuscript, the authors have attempted to address many of the concerns raised during review of the initial submission. The manuscript will benefit if the authors could further elaborate on the below mentioned aspects:

- Lines 390-391. “These effects, however, should be extrapolated cautiously because changes in the microbiota are suggested to be antibiotic-specific”. As mentioned earlier, this statement needed to be cited with appropriate examples of antibiotic classes and the resulting changes in microbiota. In the revised manuscript, the authors have only included a reference for this statement; instead a couple of lines to explain this association, along with specific examples of antibiotic classes and the microbiota impacted, would be beneficial for the readers.

- In continuation of my previous comments, can the authors propose (at least in the discussion) any alternate strategy to modulate the normal flora for the observed immunostimulatory effect without the use of antibiotics? This point remains a concern because as mentioned earlier (and agreed by the authors), antibiotic prophylaxis is not a viable option. The entire study is based on the impact of amoxicillin on gut microbiota, immune response, and subsequent Salmonella infection. The authors state “the goal is to reduce the use of antibiotics by developing novel methods to modulate the microbiome. The development of such methods is not the goal of this study. .... we do provide here the basis and justification for such studies, the concept that microbiome modulation can affect infection outcomes.” However, the concept that modulation of the microbiome can affect infection outcomes is already known. To me, the study attempts to address a mechanism for the observed effects but as the experimental layout used (amoxicillin exposure) would not be a feasible option, the manuscript will be more enriching if alternate ways to achieve this effect can at least be proposed/hypothesized.

7. PLOS authors have the option to publish the peer review history of their article (what does this mean?). If published, this will include your full peer review and any attached files.

Reviewer #1: No

Reviewer #2: No

---

## [Author Response · Author response to Decision Letter 1]

5 Mar 2020

Reviewer #1: Authors have tried to modify the manuscript entitled “Putting the microbiota to work-------------- Salmonella Typhimurium challenge in vitro” as per reviewer’s suggestions but it is still very confusing and not an exciting research paper. However, the proposed mechanism of gut modulation by antibiotic treatment in early phase of life cannot be correlated with the present study and their basis of discussion is still purely hypothetical.

Author should clarify whether they have used prophylactic or metaphylactic studies. I still feel that the article is very confusing and is not acceptable for publication.

*Respectfully, we are unsure on how to interpret these comments. This study is not a prophylactic or a metaphylactic study. Here, we described the concept of exploiting the gut microbiome as a “innate immunity” mechanism, mitigating lesions associated with Salmonella.

Thus, we were unable to identify any suggestions or alternatives to improve the manuscript based on the reviewer’s notes. The reviewer suggests that the text is “confusing” and “not exciting” without giving specific examples of what should be improved, using vague statements such as “I still feel that the article is very confusing”. 

Unfortunately, we cannot address any comments because there were no effective comments to be addressed.

Reviewer #2: In the revised version of the manuscript, the authors have attempted to address many of the concerns raised during review of the initial submission. The manuscript will benefit if the authors could further elaborate on the below mentioned aspects:

- Lines 390-391. “These effects, however, should be extrapolated cautiously because changes in the microbiota are suggested to be antibiotic-specific”. As mentioned earlier, this statement needed to be cited with appropriate examples of antibiotic classes and the resulting changes in microbiota. In the revised manuscript, the authors have only included a reference for this statement; instead a couple of lines to explain this association, along with specific examples of antibiotic classes and the microbiota impacted, would be beneficial for the readers.

*We have taken the reviewer’s suggestion into consideration and improved this section, which can be seen in the revised version of the manuscript between lines 390-398.

- In continuation of my previous comments, can the authors propose (at least in the discussion) any alternate strategy to modulate the normal flora for the observed immunostimulatory effect without the use of antibiotics? This point remains a concern because as mentioned earlier (and agreed by the authors), antibiotic prophylaxis is not a viable option. The entire study is based on the impact of amoxicillin on gut microbiota, immune response, and subsequent Salmonella infection. The authors state “the goal is to reduce the use of antibiotics by developing novel methods to modulate the microbiome. The development of such methods is not the goal of this study. .... we do provide here the basis and justification for such studies, the concept that microbiome modulation can affect infection outcomes.” However, the concept that modulation of the microbiome can affect infection outcomes is already known. To me, the study attempts to address a mechanism for the observed effects but as the experimental layout used (amoxicillin exposure) would not be a feasible option, the manuscript will be more enriching if alternate ways to achieve this effect can at least be proposed/hypothesized.

*We appreciate the reviewer’s concern and agree that, without the proper context, findings from this study could be erroneously interpreted. Thus, we have further disserted about this issue by proposing possible alternatives to the use of antibiotics to modulate the gut microbiome, including successful cases of such approach. You may find this discussion between lines 418-427 in the revised version of the manuscript.

---

## [Decision Letter · Decision Letter 2]

6 Apr 2020

Putting the microbiota to work: Epigenetic effects of early life antibiotic treatment are associated with immune-related pathways and reduced epithelial necrosis following Salmonella Typhimurium challenge in vitro.

PONE-D-19-19473R2

Dear Dr. Costa

We are pleased to inform you that your manuscript has been judged scientifically suitable for publication and will be formally accepted for publication once it complies with all outstanding technical requirements.

With kind regards,

Praveen Rishi, Ph.D., FAMI, FABMS

Academic Editor

PLOS ONE

Additional Editor Comments (optional):

Reviewers' comments:

Reviewer's Responses to Questions

**Comments to the Author**

1. If the authors have adequately addressed your comments raised in a previous round of review and you feel that this manuscript is now acceptable for publication, you may indicate that here to bypass the “Comments to the Author” section, enter your conflict of interest statement in the “Confidential to Editor” section, and submit your "Accept" recommendation.

Reviewer #2: All comments have been addressed

Reviewer #3: All comments have been addressed

2. Is the manuscript technically sound, and do the data support the conclusions?

Reviewer #2: Yes

Reviewer #3: Yes

3. Has the statistical analysis been performed appropriately and rigorously? 

Reviewer #2: Yes

Reviewer #3: No

4. Have the authors made all data underlying the findings in their manuscript fully available?

Reviewer #2: Yes

Reviewer #3: Yes

5. Is the manuscript presented in an intelligible fashion and written in standard English?

Reviewer #2: Yes

Reviewer #3: Yes

6. Review Comments to the Author

Reviewer #2: (No Response)

Reviewer #3: In the revised version of the manuscript entitled “Putting the microbiota to work------------

Salmonella Typhimurium challenge in vitro”, the authors have answered all the raised questions.

7. PLOS authors have the option to publish the peer review history of their article (what does this mean?). If published, this will include your full peer review and any attached files.

Reviewer #2: No

Reviewer #3: No

---

## [Editor Report · Acceptance letter]

13 Apr 2020

PONE-D-19-19473R2 

Putting the microbiota to work: Epigenetic effects of early life antibiotic treatment are associated with immune-related pathways and reduced epithelial necrosis following *Salmonella* Typhimurium challenge *in vitro*. 

Dear Dr. Costa:

I am pleased to inform you that your manuscript has been deemed suitable for publication in PLOS ONE. Congratulations! Your manuscript is now with our production department. 

With kind regards,

on behalf of

Prof. Praveen Rishi 

Academic Editor

PLOS ONE